# Population-Based Analysis of National Comprehensive Cancer Network (NCCN) Guideline Adherence for Patients with Anal Squamous Cell Carcinoma in California

**DOI:** 10.3390/cancers15051465

**Published:** 2023-02-25

**Authors:** Priyanka Kumar, Michael Del Rosario, Jenny Chang, Argyrios Ziogas, Mehraneh D. Jafari, Robert E. Bristow, Sora Park Tanjasiri, Jason A. Zell

**Affiliations:** 1Department of Internal Medicine, University of California, Irvine, CA 92868-3201, USA; 2Cancer and Blood Specialty Clinic, Los Alamitos, CA 90720, USA; 3Department of Surgery, Section of Colon and Rectal Surgery, Weill Cornell Medicine, New York, NY 10065, USA; 4Division of Gynecologic Oncology, Department of Obstetrics and Gynecology, University of California, Irvine, CA 92868-3201, USA; 5Department of Epidemiology & Biostatistics, University of California, Irvine, CA 92868-3201, USA; 6Division of Hematology-Oncology, Department of Medicine, University of California, Irvine, CA 92868-3201, USA; 7Chao Family Comprehensive Cancer Center, University of California, Irvine, CA 92868-3201, USA

**Keywords:** anal squamous cell carcinoma, cancer outcomes, health disparities, guideline adherence

## Abstract

**Simple Summary:**

Oncology-specific evidence-based treatment guidelines aim to improve cancer care. Less is known about the effect of guideline adherence on anal squamous cell carcinoma outcomes. Our study aimed to analyze adherence to the National Comprehensive Cancer Network treatment guidelines for anal squamous cell carcinoma in California and the associated impacts on survival. From our retrospective analysis, we found those with male sex, Medicaid insurance, and low socioeconomic status were less likely to receive adherent care. Race/ethnicity was not associated with receipt of adherent care. Adherent care was also associated with improved overall and disease-specific survival. Our study further contributes to the literature that shows guideline-adherent care improves cancer outcomes. Further efforts must therefore be made to increase guideline adherence in anal squamous cell carcinoma, especially in vulnerable populations.

**Abstract:**

Purpose: We analyzed adherence to the National Comprehensive Cancer Network treatment guidelines for anal squamous cell carcinoma in California and the associated impacts on survival. Methods: This was a retrospective study of patients in the California Cancer Registry aged 18 to 79 years with recent diagnoses of anal squamous cell carcinoma. Predefined criteria were used to determine adherence. Adjusted odds ratios and 95% confidence intervals were estimated for those receiving adherent care. Disease-specific survival (DSS) and overall survival (OS) were examined with a Cox proportional hazards model. Results: 4740 patients were analyzed. Female sex was positively associated with adherent care. Medicaid status and low socioeconomic status were negatively associated with adherent care. Non-adherent care was associated with worse OS (Adjusted HR 1.87, 95% CI = 1.66, 2.12, *p* < 0.0001). DSS was worse in patients receiving non-adherent care (Adjusted HR 1.96, 95% CI = 1.56, 2.46, *p* < 0.0001). Female sex was associated with improved DSS and OS. Black race, Medicare/Medicaid, and low socioeconomic status were associated with worse OS. Conclusions: Male patients, those with Medicaid insurance, or those with low socioeconomic status are less likely to receive adherent care. Adherent care was associated with improved DSS and OS in anal carcinoma patients.

## 1. Introduction

Although anal squamous cell carcinoma (ASCC) is a relatively rare disease, it accounts for over 1000 cancer-related deaths per year and an estimated 8200 new cancer cases [1,2]. Development of ASCC is associated with human papillomavirus (HPV), causing the disease to be more prevalent in those with known HPV risk factors, such as smokers, men who have sex with men, and those with immunodeficiencies (e.g., HIV/AIDS) [3]. The majority of ASCC cases are due to HPV, and as a result, HPV-related vaccination and screening efforts have made the disease more preventable [4,5]. However, despite this, ASCC incidence has increased over the last few decades [5,6,7,8,9]. Between 2009 and 2018, the average annual percent change (AAPC) was 0.8% among all individuals in California [10]. The rising incidence of ASCC in California is also reflected in national trends. A recent study reported that SCC rates increased 2.59% each year from 2007 to 2014 [11].

The treatment of ASCC is complex and requires a multidisciplinary approach, typically involving providers across specialties including radiology, radiation oncology, medical oncology, colorectal surgery, and gastroenterology [12]. As a result, the National Comprehensive Cancer Network (NCCN) published guidelines to help standardize care. Current NCCN guidelines recommend combination chemoradiation therapy with a pyrimidine analog (infusional 5-fluouracil, 5-FU, or oral capecitabine) and intravenous mitomycin for treatment of nonmetastatic ASCC, as this has resulted in prolonged disease-free survival: up to 80% at 5 years, especially in early-stage disease [12,13,14,15]. However, barriers to receipt of NCCN guideline directed treatment for ASCC exist [16]. Notably, socioeconomic factors, including insurance status and race, have been linked with higher risk of receiving non-adherent care [17,18]. Members of our group and others have shown that non-adherence to NCCN guidelines is associated with worse overall survival in rectal and colon cancers [19,20,21,22,23]. In particular, for ASCC, racial and income disparities have been linked to worse survival outcomes, but the exact mechanism of these disparities is unknown [24,25]. Therefore, this study aims to examine which patient socio-demographic factors affect guideline adherence and the impact of adherence on disease-specific and overall survival in ASCC.

## 2. Materials and Methods

The California Cancer Registry (CCR) is a population-based statewide cancer surveillance system that maintains data on all cancer patients diagnosed in California since 1988. Data collected from the CCR include patients’ social-demographic, tumor characteristic, treatment, and survival information. Using Surveillance, Epidemiology, and End Results (SEER) primary site codes (C210–C212, C218), we identified 9411 patients aged 18 to 79 years who were diagnosed with invasive anal cancer between 1 January 2004 and 31 December 2017 and had follow-up through 30 November 2018. Among them, histology codes for SCC (8050–8076, 8083–8084, 8123–8124) were used to narrow our original sample to only include cases of ASCC, which yielded 7837 cases. We then sequentially excluded patients who were diagnosed with ASCC as second or subsequent cancers over their lifetime (n = 1744), who were identified from death certificate or autopsy only (n = 3), and whose tumor stage was unknown (n = 1350). The final analytic sample included 4740 patients.

Besides tumor characteristics, other covariates included patient age at diagnosis, year of diagnosis, gender, race (Non-Hispanic White, Non-Hispanic Black, Asian, Other, or unknown), insurance type, and marital status. Patient social-economic status (SES) was stratified into quintiles for patients diagnosed both before and after 2006 using the Yost score and Yang index, respectively. The Yost score and Yang index are composite indices of SES within the CCR derived from principal component analysis of block group level census variables such as income, occupation, and education [26,27,28].

The institutional review board (IRB) of the University of California, Irvine (HS#2018-4735), and the State of California Health and Human Services Agency Committee for the Protection of Human Subjects (19-03-0044) approved this study as exempt.

Adherence to treatment guidelines for ASCC was based on NCCN recommendations for surgery, chemotherapy, and radiation therapy according to tumor stage (Table 1).

Descriptive statistics of our study sample were determined, and bivariate analysis of treatment adherence and clinical variables and demographics, including SES and insurance status, was conducted. A multivariate logistic regression model was used to assess the impact of social-demographic variables on the odds of receiving NCCN guideline–adherent care (AdC). Our multivariate model controlled for age at diagnosis, year of diagnosis, marital status, tumor stage, and tumor grade. To assess the potential effect of treatment adherence on survival, we performed survival analyses, with both disease-specific survival (DSS) and overall survival (OS) as outcomes. Disease-specific mortality for DSS was defined as death due to anal cancer using cause of death recorded according to International Classification of Disease criteria. Patients who died from other causes were treated as censored at the time of the event. Measurement of OS included all death during follow-up. Kaplan–Meier estimates of survival probability and log rank tests were used to perform univariate DSS and OS analyses. After verifying the proportionality assumption, we performed multivariate survival analysis using the Cox proportional hazards model. Adjusted hazard ratios (HRs) and 95% confidence intervals (CIs) were measured from our model. All data analysis for this study was conducted with SAS software, version 9.4 (SAS Institute, Cary, NC, USA). Statistical significance was defined as *p* < 0.05 for all 2-tailed tests.

## 3. Results

We sampled 4740 patients with a diagnosis of ASCC (Appendix A). The largest group of patients in our study were greater than 65 years in age (37.5% or 11,779 patients). There was a higher prevalence of ASCC after 2010 than before (66.4% or 3148). Most patients in our sample were female, non-Hispanic white, single/otherwise not married/unknown marital status, had managed care insurance, and were of higher-middle SES (Appendix A). Of the patients sampled, 20.1% were stage I, 35.5% were stage II, 35.6% were stage III, and 8.8% were stage IV. The majority of patients had grade II or moderately well differentiated tumor differentiation (31.1%) (Appendix A).

Overall, 3814 patients (80.5%) received NCCN guideline–adherent care (AdC) (Appendix A). Most patients in every age group of our sample received AdC. Our logistic regression model found race/ethnicity status did not affect the likelihood of receiving adherent care. Age at cancer diagnosis was found to be associated with adherence (*p* < 0.0001). Younger patients had higher odds of receiving AdC as compared to older patients. Women had higher odds of receiving AdC when compared to males (OR 1.38, 95% CI = 1.18, 1.62, *p* < 0.0001). Patients with no insurance or unknown insurance status had lower odds of receiving AdC (OR 0.60, 95% CI = 0.43, 0.85, *p* = 0.0041). Patients with the lowest SES had lower odds of receiving AdC as compared to those in the highest SES group (OR 0.65, 95% CI = 0.50, 0.83, *p* = 0.0007). Married patients had higher odds of receiving AdC as compared to those with single/separated/divorced/widowed/unknown marital status (OR 1.35, 95% CI = 1.14, 1.58, *p* = 0.0004). Patients with tumor stages II through IV had higher odds of receiving AdC as compared to patients with stage I tumors (stage II: OR 1.82, 95% CI = 1.50, 2.20, *p* < 0.0001; stage III: OR 2.64, 95% CI = 2.15, 3.24, *p* < 0.0001; stage IV: OR 1.65, 95% CI = 1.24, 2.20, *p* < 0.0001). Patients with grade III or poorly differentiated tumors had higher odds of receiving AdC than those with grade I or well-differentiated tumors (OR 1.52, 95% CI = 1.15, 2.00, *p* = 0.0032) (Appendix A).

In adjusted survival analysis using the Cox proportional-hazards model, we observed that DSS was significantly worse in patients who received non-adherent care (non-AdC) as compared to those who received AdC (HR 1.96, 95% CI = 1.56, 2.46, *p* < 0.0001) (Appendix A). In this model, lower-middle SES was associated with worse DSS, and female sex was associated with improved DSS. With log rank tests, there was no significant difference in DSS between adherent and non-adherent groups in early-stage (Stage I or Stage II) disease (Appendix A). However, DSS was significantly worse in patients with late-stage (Stage III or Stage IV) disease who received non-adherent care as compared to those who received adherent care with late-stage disease (Log rank *p* < 0.0001) (Appendix A).

OS was also worse in those receiving non-AdC as compared to those receiving AdC (HR 1.87, 95% CI = 1.66, 2.12, *p* < 0.0001) (Appendix A). Female sex was independently associated with improved OS, with HR 0.60 (95% CI = 0.54, 0.67, *p* < 0.0001). Non-Hispanic Black patients were observed to have worse OS as compared to non-Hispanic White patients after adjustment for relevant clinical variables, with HR = 1.61 (95% CI = 1.33, 1.94, *p* < 0.0001). Medicare and Medicaid patients had worse OS as compared to managed care (Appendix A). Patients in the lowest SES group had worse OS, with HR 1.66 without controlling for adherence status (95% CI = 1.38, 1.99, *p* < 0.0001) and HR 1.56 controlling for adherence status (95% CI = 1.30, 1.88, *p* < 0.0001). Lower-middle SES was also associated with worse OS, with HR 1.49 without controlling for adherence status (95% CI = 1.26, 1.78, *p* < 0.0001) and HR 1.48 controlling for adherence status (95% CI = 1.25, 1.76, *p* < 0.0001). With log rank tests, OS was worse in those who received non-adherent care as compared to adherent care in patients with early-stage (Stage I or Stage II) and late-stage (Stage III or Stage IV) disease (Appendix A).

## 4. Discussion

In this retrospective population-based analysis of anal squamous cell cancer cases identified from the CCR, we observed relatively high rates of compliance, with 80.5% of patients receiving AdC. Race/ethnicity was not significantly associated with receipt of AdC. Factors associated with receipt of AdC included female sex, insurance status, SES, and marital status. Individuals in the lowest SES group and those without insurance/with unknown insurance status were less likely to receive adherent care. We observed both DSS and OS were worse in patients receiving non-AdC as compared to those receiving AdC in our initial model.

Compliance with NCCN guideline adherence was relatively high (exceeding 80%) in our study, in contrast with other studies that have reported lower compliance rates for ASCC and across different cancer types. For instance, Lee et al. found only 26.7% of patients with early-stage epithelial ovarian cancer received guideline-adherent surgical staging [29]. Similarly, Thiels et al. and Visser et al. found 30.1% of patients with gastric cancer and 34.5% of patients with pancreatic cancer, respectively, received guideline-adherent care [30,31]. Rates of compliance have ranged from as low as around 30% to higher rates of compliance from nearly 60–80% compliance, depending on cancer type and region of practice [16,23,32,33]. One potential explanation for the high proportion of NCCN AdC observed in our study compared with reports in other cancers is that, in general, there is greater consensus on the guidelines for the treatment of ASCC, supported by clinical trial–based research. Given the relationship between NCCN compliance and improved survival across different cancers, our results may suggest the need to improve guideline compliance and conduct more research on the factors affecting compliance.

Patient age at diagnosis was also associated with receipt of NCCN guideline–adherent care. Particularly, we found younger patients had higher odds of receiving adherent care. This finding is consistent with similar studies in the literature. A retrospective single-center study of patients with epithelial ovarian cancer by Erickson et al. found that women who received adherent care were on average younger as compared to those who received non-adherent care (61.9 years vs. 69.0 years, *p* = 0.009) [34]. Similarly, another study by Visser et al. observed pancreatic cancer patients older than 65 years were less likely to receive compliant care after adjusting for patient and hospital-related factors [31]. A possible explanation for younger patients being more likely to receive compliant care may be due to age-related comorbidities or lower functional status. For example, older patients may have more comorbidities or lower functional status that prevent them from receiving standard of care therapies for their disease, including chemotherapy, which may be more toxic and therefore contraindicated.

Race and ethnicity did not impact receipt of NCCN-adherent care in our study. In addition, across both AdC and non-AdC groups, race/ethnicity was not found to significantly impact DSS. However, non-Hispanic Black patients in both AdC and non-AdC groups experienced worse OS as compared to non-Hispanic White patients. Our findings contrast with those of other studies, which have typically found minority race to have worse cancer outcomes, including adherence to guidelines, delays in treatment, and survival [24,35,36,37,38]. A study by Patel et al. showed non-white race was associated with worse survival and reduced receipt of chemoradiation treatment in ASCC patients. The authors suggested these associations showing race-related disparities may be mediated by lack of access to treatment [38]. Ahmad et al. demonstrated that minority race predicted worse relapse-free survival (RFS) and OS in a multivariate analysis, suggesting race may mediate outcomes independently from economic factors, such as SES and/or insurance status [37].

We found ASCC patients with no insurance or unknown insurance status were less likely to receive AdC. Our study also found patients in the lowest SES groups were less likely to receive AdC as compared to those in the highest SES group. The negative relationship between uninsured status, lower SES, and receipt of AdC is expected and likely explained by a lack of access to resources in these vulnerable populations. Specifically, financial toxicity has been associated with nonadherence to treatment plans and less access to cancer care, including supportive treatments and clinical trials, through both high direct and indirect costs of care [39,40]. Interestingly, our findings conflict with the results of a recent study by Patel et al., which found no relationship between lower SES and uninsured status and receipt of chemoradiation therapy amongst ASCC patients in the SEER database. However, they found that Medicaid status independently predicted reduced receipt of chemoradiation therapy and noted that uninsured patients had worse cancer survival. In their study, lower income was also associated with worse survival but not with receipt of chemoradiation therapy. The authors explain that the discrepancies between survival and receipt of chemotherapy and radiation therapy in uninsured and low-income populations and might be due to worse treatment adherence [38].

Differences in our study and the results of Patel et al. might be due to environmental differences in sample populations between the national SEER database and the CCR. Pricolo et al. found lower rates of guideline compliance in ASCC patients treated at community cancer centers as compared to academic centers [41]. It is therefore possible that non-patient-related factors, such as regional practice variation, may affect rates of compliance in patients in both the SEER and CCR databases.

Patients with tumor stages II through IV had higher odds of receiving adherent care as compared to patients with stage I disease. In addition, patients with grade III or poorly differentiated tumors were also found to have higher odds of compliant care as compared to those with grade I or well-differentiated tumors. Our results generally contrast with studies on other types of cancer, which have found patients with greater tumor stage had a greater risk of receiving noncompliant care [17,34]. For instance, Hines et al. observed colorectal cancer patients with stage II through IV tumors were at greater risk for receiving noncompliant care, attributing this to the additional need for neoadjuvant and adjuvant chemotherapy in stage II and III disease [17].

However, our results are consistent with other studies on ASCC, including those from Bian et al. and Kole et al. [16,42]. Bian et al. performed a retrospective analysis of ASCC patients with stage II and III disease in the National Cancer Database (NCD) and found patients with stage III (as compared to lower stage II disease) and poorly differentiated or undifferentiated tumors were more likely to receive standard therapy [42]. Kole et al. also noticed a similar relationship; ASCC patients with low-grade tumors were significantly less likely to receive adherent care in chemoradiation therapy as compared to those with intermediate/high grade tumors (AOR 0.24, 95% CI 0.14, 0.29, *p* < 0.001) [16]. Patients with lower stage and grade tumors may be more likely to receive non-adherent care due to the tendency to treat patients with less risk of recurrence less “aggressively”. Findings by Kole et al. support this idea, given the authors noted the strongest predictors of noncompliance in their study included low tumor grade and small tumor size, both of which correspond to better prognosis [16]. In Stage I disease in particular, lower guideline compliance may also occur due to the variety of treatment options, including chemoradiation or surgery alone [43]. Patients may receive radiation after surgery with or without chemotherapy due to incomplete knowledge of tumor involvement/risk factors prior to surgery, which may occur in cases of anal margin carcinoma.

Both DSS and OS were worse in ASCC patients receiving non-AdC as compared to patients receiving AdC. This result is consistent with the literature, which has shown a strong relationship between adherence to NCCN guidelines and improved outcomes, including survival across different cancers such as ovarian cancer, colorectal cancer, melanoma, pancreatic cancer, and gastric cancer [22,29,31,33,44,45]. Our findings contribute to the body of literature that suggests NCCN compliance may be a measure of quality of care. It therefore remains a priority to study the factors affecting compliance to develop targeted approaches to increase access to care for vulnerable populations.

There are several limitations to our retrospective population-based study. One of the major limitations of our study was not having detailed clinical data on patient treatment, including specific drugs utilized or radiation treatment dosing, treatment duration, and timeliness of treatment completion. As such, it is difficult to ascertain which patients may have truly received complete treatment in strict adherence to NCCN guidelines. In addition, CCR lacks information on comorbidities such as HIV, immunosuppression, HPV status, and smoking status, including well-known risk factors for ASCC that may influence treatment outcomes and overall survival [24,38]. For example, in recent studies, HIV-positive status was shown to affect treatment outcomes, such as experiencing more acute treatment-related toxic effects [46,47], but interestingly it did not affect cancer-specific and all-cause mortality [46]. Another study by Hammad et al. noted that HIV-positive patients were less likely to receive full-dose chemoradiation therapy, implying lower guideline adherence [48]. The relationship between guideline adherence and HIV-positive status might therefore be related to increased treatment-related toxicities in this population. Like HIV, HPV status has been known to affect ASCC outcomes, including survival [24,38]. A meta-analysis of epidemiological studies by Sun et al. revealed HPV+/p16+ cancers were associated with improved DSS and OS as compared to HPV−/p16+ or HPV+/p16− cancers [49]. Another meta-analysis of individual patient data by Obermueller et al. also showed patients with HPV+/p16+ cancers had improved OS as compared to individuals with HPV−/p16− disease [50]. An additional limitation of the CCR database is that it lacks data on other general medical comorbidities involving frailty and/or functional status. Poor functional status and increased frailty, especially in elderly populations with higher risk of treatment-related toxicities, ultimately affect adherence. We also recognize limitations in our measurement of SES. The Yost score and Yang index provide a useful measurement of SES in larger, homogenous areas. However, many California counties contain a heterogenous mix of higher and lower SES neighborhoods, which may reduce the applicability of this measurement. Lastly, our study was limited by the exclusion of patients without known tumor stage, which may have affected our results.

Despite its limitations, our study has unique strengths. Data from the CCR are limited to one state, California, which may limit external validity. However, as a population-based study of a geographically contiguous region with an underlying population approaching 40,000,000 residents [51], our study offers inherent strengths in internal validity as compared to other more widely available cancer registries, such as the Surveillance, Epidemiology, and End Results Program (SEER), a collection of smaller U.S.-based cancer registries in various areas of the United States. Another unique strength of our study is the ability to analyze a large and diverse dataset, which includes populations that have historically been underrepresented in outcomes research (e.g., Hispanics, Blacks, and Asian Americans). While contributing to the current literature on underrepresented minorities in outcomes research, our study re-emphasizes the need for further research in this field [52,53,54,55].

## 5. Conclusions

Our retrospective population-based study found NCCN guideline adherence was associated with both increased disease-specific and overall survival in patients with ASCC. In addition, female sex, SES, insurance status, and marital status were all associated with adherence. Race/ethnicity was not found to significantly affect receipt of AdC, which may have been due to relatively small sample size. Our findings highlight the need for greater delivery of guideline adherent care to vulnerable populations to improve outcomes for all ASCC patients.

## Figures and Tables

**Table 1 cancers-15-01465-t001:** Definition of NCCN guideline–adherent care.

Anal Cancer Treatment Adherence		
Adherent treatment		
Surgery	Chemotherapy	Radiation therapy
Stage I (T1, N0 Well or Mod differentiated or T2, N0, which does not involve sphincter)		
Local Excision	Yes/No	Yes/No
Stage I, II, III		
None	Yes	Yes
Stage IV		
Yes/No	Yes	Yes/No

## Data Availability

Publicly available datasets were analyzed for the purposes of this study. This data can be found here: https://www.ccrcal.org/retrieve-data/.

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
