# Peer review of "Population-Based Analysis of National Comprehensive Cancer Network (NCCN) Guideline Adherence for Patients with Anal Squamous Cell Carcinoma in California"

_cancers, 2023, doi:10.3390/cancers15051465_

Round 1

Reviewer 1 Report

This is a population-based retrospective analysis on the NCCN adherence care and the impact of socio-economic factors on anal squamous cell carcinoma (ASCC) oncological outcome, based on a national database.

The most attractive item of the paper is the relevant number of patients included (4740 patients), affected by a relatively rare cancer.

Moreover, the statistical analysis and the methodology is correct and well-conducted.

However, as the authors themselves stated, there are some limitations in this study that negatively affect the final view.

The topic is interesting, as the adherence of care to national guidelines and factors affecting it, is a very debated argument; infact, it has been proven that socio-economic factors play a relevant role in adherent of care and impact on survival.

Many patient data are lacking in the database, such as comorbidities, oncological follow up, type of treatment and treatment duration, which are strongly related to oncological outcomes. 

This factor impairs the relevance of the paper. 

Also, there are many papers reporting about the same topic, using similar sources . Although the database used is large and complex, it reflects the social and economic impact factors of a single reality, California.

Author Response

Reviewer #2:

- This is a population-based retrospective analysis on the NCCN adherence care and the impact of socio-economic factors on anal squamous cell carcinoma (ASCC) oncological outcome, based on a national database.

-The most attractive item of the paper is the relevant number of patients included (4740 patients), affected by a relatively rare cancer. Moreover, the statistical analysis and the methodology is correct and well-conducted.

Author response: Thank you!

- However, as the authors themselves stated, there are some limitations in this study that negatively affect the final view. The topic is interesting, as the adherence of care to national guidelines and factors affecting it, is a very debated argument; in fact, it has been proven that socio-economic factors play a relevant role in adherent of care and impact on survival.

- Many patient data are lacking in the database, such as comorbidities, oncological follow up, type of treatment and treatment duration, which are strongly related to oncological outcomes. 

-This factor impairs the relevance of the paper. 

Author response: Thank you for bringing this to our attention. Information on comorbid conditions, especially HIV and/or HPV status, are missing from the California Cancer Registry (CCR) and are a notable limitation in working with the database. We have revised our manuscript to include an explanation of this limitation in our discussion section (pp. 14-15). The authors, however, would like to add that our disease-specific survival analysis is less affected by comorbid conditions as compared to overall survival.

The authors would like to clarify the reviewer’s definition of “oncological follow up” in the above statement. The CCR includes information on both patient vital status and cause of death, which allowed calculation of disease-specific survival and overall survival. As a large state-wide database, the CCR does not include clinical/EMR documentation from individual patients whose data were compiled. Does “oncological follow up” mean to refer to treatment duration, treatment completion, and/or compliance with treatment(s)?

Thank you for this feedback. The CCR does not have information on specific drug names or radiation treatment dosing; however, the registry does have information on the broad categories of treatment used. We analyzed NCCN guideline adherence by this information. We have revised our discussion section accordingly to include a statement that clarifies this point/limitation of the database (p. 14).

- Also, there are many papers reporting about the same topic, using similar sources. Although the database used is large and complex, it reflects the social and economic impact factors of a single reality, California.

Author response: The CCR reports on data from a single state, which may unfortunately limit the external validity of our study. We have newly included an explanation of this limitation in our discussion (p. 15). In addition, the authors would like to highlight some unique strengths of our manuscript. The large number of patients (and large number of patients specifically with anal cancer) within the registry offers increased internal validity. Furthermore, another unique aspect of our manuscript is its ability to analyze a large and racially diverse dataset for populations that are often underrepresented in outcomes research (e.g., Hispanics, Blacks, and Asian Americans). We revised our manuscript to include further explanation of these strengths in our discussion section (p. 16). We have also updated our references to include studies that highlight the need for more representation of different minority groups in outcomes research (p. 29).

Reviewer 2 Report

Dear authors

Congratulations for the work reporting an important topic in the management of all cancers.

The acquisition of data from the registry has allowed you to report a very large number of cases on a pathology that remains a rare tumour. The epidemiological changes occurred in recent years also emerge from these data.

I report my comments below.

- The overall adherence to the guidelines is good considering that ASCC is a rare tumour. It would be interesting to include the experience of the centers in which the patients were treated as well as the presence of dedicated multidisciplinary teams and the possible significance of these factors in adherence to guidelines

- The greater compliance with the guidelines in the treatment of young patients, as reported in the discussion could derive from comorbidities and frailty of elderly patients and the risk of chemo-radiation related  toxicity, the absence of these data makes this result less useful.

- In assessing the prognosis as well as in the therapeutic choice, current clinical practice cannot disregard the assessment of the patients’ HIV status and the correlation between the tumor and the HPV infection. HIV infection and its status could also indirectly correlate with socio-economic status. The lack of these data is a very important limitation and I recommend dedicating more space to this aspect in the discussion.

- the lower adherence to the guidelines in stage I could depend on the different thearpeutic options previded by the guidelines, moreover as reported by some studies in the literature, sometimes the diagnosis is not recognized until the surgical removal of the lesion, especially anal margin carcinoma cases. This aspect could be included in the discussion by emphasizing that many patients undergo radiotherapy with or without chemotherapy after surgery because the risk factors were not known before surgery.

Author Response

Reviewer #1:

Congratulations for the work reporting an important topic in the management of all cancers.

The acquisition of data from the registry has allowed you to report a very large number of cases on a pathology that remains a rare tumour. The epidemiological changes occurred in recent years also emerge from these data.

Author response: Thank you!

I report my comments below.

- The overall adherence to the guidelines is good considering that ASCC is a rare tumour. It would be interesting to include the experience of the centers in which the patients were treated as well as the presence of dedicated multidisciplinary teams and the possible significance of these factors in adherence to guidelines

Author response: The authors agree this would be an interesting factor to look at in relationship to adherence. Unfortunately, the CCR includes limited and unidentifiable information at the institutional level where patients were first evaluated and/or treated. 

- The greater compliance with the guidelines in the treatment of young patients, as reported in the discussion could derive from comorbidities and frailty of elderly patients and the risk of chemo-radiation related toxicity, the absence of these data makes this result less useful. 

Author response: Thank you for this feedback. We have more clearly highlighted this limitation (the lack of information on comorbidities in the CCR) in our discussion section (p. 15).

- In assessing the prognosis as well as in the therapeutic choice, current clinical practice cannot disregard the assessment of the patients’ HIV status and the correlation between the tumor and the HPV infection. HIV infection and its status could also indirectly correlate with socio-economic status. The lack of these data is a very important limitation and I recommend dedicating more space to this aspect in the discussion.

Author response: Thank you! We agree that this represents an important limitation to our manuscript and have revised our discussion to include more discussion of this (pp. 14-15).

- the lower adherence to the guidelines in stage I could depend on the different therapeutic options provided by the guidelines, moreover, as reported by some studies in the literature, sometimes the diagnosis is not recognized until the surgical removal of the lesion, especially anal margin carcinoma cases. This aspect could be included in the discussion by emphasizing that many patients undergo radiotherapy with or without chemotherapy after surgery because the risk factors were not known before surgery.

Author response:  This is valuable feedback, and the authors appreciate it. We have addended our discussion to reflect this point (p. 13).

Round 2

Reviewer 1 Report

Retrospective study. 
the topic is not attractive and has been already discussed in other papers.

Methodology is good 

discussion is redundant

Author Response

Author response: Thank you for this feedback! In response to the reviewer’s comment that there was redundancy in the discussion section, we have revised the discussion accordingly to remove any redundant sections (pp. 4-7).